# Dietary Biodiversity and Diet Quality in Dutch Adults

**DOI:** 10.3390/nu16142189

**Published:** 2024-07-09

**Authors:** Rosalie E. Bakker, Vera S. Booij, Corné van Dooren, Mary Nicolaou, Ingeborg A. Brouwer, Margreet R. Olthof

**Affiliations:** 1Department of Health Sciences, Faculty of Science, Amsterdam Public Health Research Institute, Vrije Universiteit Amsterdam, 1081 HV Amsterdam, The Netherlands; 2WWF-NL, 3708 JB Zeist, The Netherlands; 3Department of Public and Occupational Health, Amsterdam UMC, University of Amsterdam, 1007 MB Amsterdam, The Netherlands

**Keywords:** biodiversity, food biodiversity, dietary diversity, species richness, diet quality

## Abstract

Dietary biodiversity, defined as the variety of consumed plants, animals and other organisms, can be measured by dietary species richness (DSR). This study investigated associations between DSR and diet quality in Dutch adults. Dietary intake data of 2078 Dutch participants, aged 19 to 79 years, were collected by the Dutch National Food Consumption Survey between 2012 and 2016 via two non-consecutive 24-h dietary recalls. DSR scores were calculated based on the total count of unique species consumed per individual over the two measurement days. An overall DSR score and separate scores for fruit and vegetable species consumption were calculated. The Dutch Healthy Diet index 2015 (DHD15-index) was used to measure diet quality. Linear regression analyses were performed to investigate associations between DSR scores and DHD15-index. Analyses were stratified by age and adjusted for relevant confounders. In total, 157 unique species were identified within the investigated food groups. On average, individuals consumed 13 unique species over two days (SD 4.55). For every additional species consumed, the DHD15-index increased by 1.40 points (95%CI 1.25–1.55). Associations between DSR and DHD15-index were higher in younger adults. DSR fruit showed the strongest associations with DHD15-index (β 4.01 [95%CI 3.65–4.38]). Higher DSR scores are associated with higher diet quality in Dutch adults. These newly developed DSR scores create opportunities for further research to explore the implications of dietary biodiversity in Western diets on health and related outcomes.

## 1. Introduction

Low quality diets are an important contributor to the global burden of disease and were responsible for almost 20% of the total deaths in 2016 worldwide [1,2,3]. Over one billion people suffer from micronutrient deficiencies and over 800 million people are undernourished [4,5]. Dietary behavior, such as high caloric intake, alcohol intake and insufficient intake of fruits, vegetables and fiber-rich products, collectively contributes to an increasing burden of disease [1,6].

Globally, human diets have increasingly become more uniform, marked by increased (ultra-) processed food consumption and a decline in diversity [7,8,9]. While thousands of edible species are available, human diets are predominantly characterized by four crops: rice, potatoes, wheat and maize, covering half of the energy need in global dietary intake [10,11]. Dietary diversity has been suggested to result in adequate nutrient intakes and most national and international dietary guidelines encourage people to adopt a varied diet; however, the link and mechanisms between dietary diversity and human health remains uncertain [12,13,14,15,16,17,18].

Human diets and the diversity or uniformity are also related to nature. Diversity in diets is linked via biodiversity in agricultural practices to wild biodiversity and nature [10,11,19,20,21]. Higher dietary diversity could lower pressure on certain species commonly consumed in the current diet [22,23]. Previous research showed associations between dietary diversity scores and diversity in agricultural production in low- and middle-income countries [11]. However, the link between agricultural biodiversity and diet quality has not been thoroughly investigated yet, especially not in Western countries [19]. The concept of biodiversity in dietary patterns could provide valuable insights or contribute solutions to counter biodiversity loss and promote healthy diets in line with the Sustainable Development Goals established by the United Nations [11,24].

Dietary biodiversity encompasses the variety of animal, plant or other species used for consumption. Dietary species richness (DSR) has been recommended as the most appropriate measure to capture dietary biodiversity, or species richness, in human diets [25]. Lachat et al. (2018) established that higher DSR is associated with higher micronutrient adequacy, an aspect of diet quality, in women and children in low- and middle-income countries [25]. However, it is uncertain if this relationship also exists for men and if it occurs in high-income countries, where food consumption consists of more processed foods bought from supermarkets. Hanley-Cook et al. (2021) assessed DSR in Western dietary patterns and found that higher DSR scores were associated with lower mortality risks in European populations [26]. However, uncertainty remains regarding the pathways through which this association operates and whether diet quality serves as a connecting factor between dietary biodiversity and health outcomes. Moreover, it remains unclear if a relationship between dietary biodiversity and diet quality exists in Western populations. Therefore, this study aims to investigate whether DSR and diet quality are associated in Dutch adults (aged 19–79 years).

## 2. Methods

### 2.1. Study Population and Design

Data for this study were retrieved from the Dutch National Food Consumption Survey (DNFCS) 2012–2016 carried out by the National Institute for Public Health and the Environment of the Netherlands [27]. The aim of the survey is to obtain insight into the diets of Dutch children and adults and other additional lifestyle factors.

Food consumption data were collected among the general Dutch population between 2012 and 2016. Participants needed to be between 1 and 79 years old. Pregnant or lactating women and institutionalized people were not eligible for participation. Furthermore, participants needed to have an adequate understanding of the Dutch language. Every participant completed an age specific general questionnaire, which covered various personal and lifestyle factors. DNFCS 2012–2016 is a national representative sample. A detailed description about the development and execution of this survey can be found elsewhere [27]. A total of 4313 participants’ data were gathered from the DNFCS 2012–2016. For this study, only data from adults aged 19 years or older were included (N = 2078).

### 2.2. Dietary Assessment

All participants completed two non-consecutive 24-h recall interviews by phone or face to face to properly collect dietary intake data. The interval between the two interviews was approximately four weeks and all interviews were spread equally over all days of the week and the four seasons at population level. The interviews were led by trained dietitians who were familiar with the GloboDiet system. GloboDiet (IARC©; former EPIC-Soft) is a computer-controlled interview software that directly digitally stores the interview information and can link it to relevant food composition data [28]. Food composition, nutrients and energy (caloric) intake data originated from the Dutch Food Composition Database [29].

### 2.3. Dutch Healthy Diet Index 2015

Diet quality was measured by the Dutch Healthy Diet index 2015 (DHD15-index), which was developed and validated by Looman et al. (2017) [30]. The DHD15-index can be used as an indicator of the healthiness of the diet of Dutch respondents, since it measures the adherence to the Dutch dietary guidelines for a healthy diet according to a ranking system [13,30]. The DHD15-index consists of fifteen components, namely, vegetables, fruits, wholegrain products, legumes, nuts, dairy, fish, tea, fats and oils, coffee, red meat, processed meat, sweetened beverages and fruit juices, alcohol and sodium. For all components, average food consumption was determined over two days and a score between 0 (no adherence) and 10 points (complete adherence) was awarded. Originally, the range of the DHD15-index is between 0 (no adherence) and 150 (complete adherence). However, in this study, the component ‘coffee’ was excluded since there was no data in the DNFCS that distinguishes if the consumed coffee was (un)filtered, which determines the score in this component. Therefore, the highest possible DHD15-index in this study was 140. Per participant, a mean DHD15-index score was calculated over two measurement days. A further detailed explanation of the calculation of the DHD15-index can be found elsewhere [31].

### 2.4. Dietary Biodiversity Calculation

Dietary biodiversity was measured by dietary species richness (DSR), previously recommended as the most appropriate measure for dietary biodiversity in dietary intake studies [25]. Lachat et al. (2017) highlighted that DSR showed stronger and more consistent associations with diet quality indicators than other relevant indicators, such as Simpson’s diversity index (D) and functional diversity (FD). Moreover, DSR has also been used in Western dietary intake studies [25,26]. DSR measures diversity within food groups since it makes distinctions on species level, unlike other diet diversity scores that primarily measure diversity between food groups [12,25,26,32,33].

In this study, DSR was measured by the total count of species consumed per individual over two measurement days. Data of two 24-h dietary recalls of the DNFCS were used in order to calculate an individuals’ DSR, regardless of the quantity (grams or calories) of the consumed species, different consumption moments of species or consumption in different forms.

To calculate individual DSR scores, first, all consumed species (plant, animal or fungi) within the population needed to be identified. The data consisted of food and drink descriptions corresponding to the codes used in the national Dutch Food Composition Database [29]. Based on common food names, a classification of species was made. Two experts with botanical knowledge were consulted to check the species classification for fruits and vegetables developed by the researchers. Food registrations or products that consisted out of a combination of species (e.g., tinned tropical fruit cocktail, vegetable mix Italian style, etc.) were identified and classified as different species. Species of composite dishes were determined according to the information available in the national Dutch Food Composition Database. In case this information was lacking, standard recipes of comparable products available in Dutch supermarkets were used to determine the different species in one product [34].

After determining all species in the database (Appendix A), a DSR score per participant was calculated. The overall DSR score included the food groups: fruits, vegetables, nuts and seeds, legumes, tubers, grains, meat, fish and dairy. Unprocessed foods, cut-up, tinned, jarred and dried foods and/or (deep)frozen foods within these food groups were included when computing the DSR scores. Alcoholic and non-alcoholic beverages, savory snacks, sugar, confectionery, oils, fats, condiments, spices, sauces and processed meat substitutes and other highly processed foods were excluded from the overall DSR calculation. In addition, separate DSR scores for fruit and vegetable species consumption were calculated. Some species were included in the DSR vegetables category even though, from a biological perspective, they are not vegetables but fall under legumes or tubers (Appendix A); this classification is also in line with the national dietary guidelines. More information about the included food groups in the DSR scores can be found in Appendix A.

Individual DSR scores were aggregated in SPSS based on the sum of the total unique species consumed. Duplicates in consumed species were discarded in the DSR scores. Therefore, different consumption moments of species and the amount or consumption of different forms of one species did not lead to a higher DSR score. Participants that did not consume any (fruit or vegetable) species scored a DSR of ‘0’ (zero).

### 2.5. Covariates

Demographic covariates included sex, educational level, energy intake (kilocalories) and age. Educational level was derived from eight detailed educational stages: low (no education, primary education, lower vocational education, advanced elementary education), middle (intermediate vocational education, higher secondary education) and high (higher vocational education/university). In the analyses, educational level consisted of only three categories, i.e., low, middle and high educational level. Age was an a priori selected effect modifier and divided into four different age groups (19–30 years, 31–50 years, 51–64 years, ≥65 years).

### 2.6. Statistical Analysis

After computing DSR scores, summary and descriptive statistics were calculated. Continuous variables were summarized using means and standard deviations, while categorical variables were expressed as frequencies and percentages. DSR scores, representing absolute numbers of consumed species, were also presented using medians and interquartile ranges due to the discrete nature of the data. Descriptive data were presented separate for different age groups and sexes.

Beta coefficients and 95% confidence intervals for the association between DSR and DHD15-index were obtained via linear regression analyses. Analyses were stratified by the four different age groups. In total, three different models were obtained via the linear regression analysis. Model 1 was adjusted for sex and educational level. Model 2 was additionally adjusted for mean total intake in kilocalories. For DSR fruit and DSR vegetables, a third model was applied which was also adjusted for the quantity of fruits or vegetables consumed in grams. Post hoc regression analyses were performed to assess the influence of seasonality on the association between DSR and DHD15-index. *p*-values of <0.05 were considered as significant. All analyses were performed in IBM SPSS Statistics Version 28.

## 3. Results

### 3.1. Descriptives

In total, 2078 Dutch adults aged 19 to 79 years old were included (Table 1). Across all age groups, most participants had a middle educational level. Only the older population (≥65 years) included more participants with a low educational level than with a high or middle educational level. Women reported a higher consumption of fruit (in grams) than men. Young women (19–30 years) ate fewer grams of vegetables than young men (19–30 years). Older adults (≥65 years) reported consuming more fruits and vegetables in grams over two days than younger adults. The DHD15-index scores increased slightly with age. Women reported higher DHD15-index scores than men; the highest scores were observed in women ≥65 years old.

### 3.2. DSR Scores

In total, 157 unique species were identified. Cow (*Bos taurus*), wheat (*Triticum aestivum*), onions (*Allium cepa*) and cabbages (*Brassica oleracea*) were the most frequently consumed species. The overall DSR scores per participant ranged between 2 and 33 over two measurement days (Figure 1). Most participants consumed one or two different fruit species over two measurement days. In total, 22% of the participants did not consume any fruit species over two measurement days and therefore had a DSR fruit score of ‘0’ [zero] (DSR fruit min.0–max.12). For vegetables, most participants consumed between two to four different vegetable species over two measurement days. Only 2% of the sample did not consume any vegetable species over two measurement days and therefore had a DSR vegetables score of ‘0’ [zero] (DSR vegetables min.0–max.15).

### 3.3. Dietary Biodiversity and Diet Quality

DSR overall scores were significantly positively associated with DHD15-index in all age groups (Table 2). On total population level, all models showed significant positive associations between DSR scores and DHD15-index. The association between DSR vegetables and DHD15-index was no longer statistically significant when adjusted for the quantity of consumed vegetables (in grams). The reported associations between DSR scores (overall, fruit and vegetables) and DHD15-index were more pronounced after adjustment for mean energy intake (kcal/day) (Model 2). Furthermore, the models show higher associations between DSR scores and DHD15-index in younger adults (19–30 years) compared with older adults. DSR fruit was significantly associated with the DHD15-index (β 4.01 [95%CI 3.65–4.38]) and showed overall stronger associations than DSR overall and DSR vegetables in all age groups. When adjusting for the quantity of consumed fruits over two days (in grams), the observed associations between DSR fruit and DHD15-index markedly attenuated and the association in adults aged 31–50 years became non-significant. DSR vegetables was significantly associated with DHD15-index (β 0.84 [95%CI 0.61–1.08]), except for adults ≥65 years (β −0.34 [95%CI −0.09–0.77]). When adjusting for the quantity of consumed vegetables over two days (in grams), all associations between DSR vegetables and DHD15-index attenuated and became non-significant. Post hoc analyses showed that seasonality was not a relevant confounding factor in the association between DSR and DHD15-index. No substantial changes in regression coefficients were found.

## 4. Discussion

This study showed that higher DSR scores were associated with higher DHD15-index scores in almost all age groups of the Dutch population. Only the association between DSR vegetables and DHD15-index for older adults (≥65 years old) was non-significant. Strongest associations between DSR and DHD15-index were found in younger adults.

The findings of this exploratory research are in line with previous studies; however, research populations differ. Lachat et al. (2018) found that higher DSR scores were associated with better micronutrient adequacy, which is an aspect of diet quality, in women and children in low- and middle-income countries (LMIC’s) [25]. Hanley-Cook et al. (2021) investigated nine different European cohorts in their study and showed that higher DSR scores were associated with lower mortality rates even when accounting for other lifestyle and dietary risk factors [26]. Nevertheless, the study of Hanley-Cook et al. is the only study investigating the association between dietary biodiversity and health outcomes in Western populations. A previous Dutch study reported associations between higher DHD15-index scores and lower mortality rates, affirming the link between diet quality and health outcomes [35]. Furthermore, McDonald et al. (2018) discovered that consuming a greater variety of plant species is associated with higher microbial diversity in the human gut, which is often linked to better health. Additionally, the study found that participants with a more diverse plant-based diet had lower levels of certain antibiotic resistance genes [36]. This suggests that dietary biodiversity may support a healthier microbiome and reduce the prevalence of antibiotic-resistant bacteria. Overall, the findings highlight the potential health benefits of consuming a wide range of plant species. To date, the possible interplay among dietary biodiversity, diet quality and health outcomes in Western populations has not yet been extensively investigated.

Strengths of this study include the size of this sample (N = 2078) and the national representativeness of the sample. Also, seasonality and its influence on the diet was taken into account since data was collected continuously from November 2012 through January 2017 [27]. The dietary intake data used in this study were of high quality since they were obtained by trained dietitians and a number of checks were conducted to ensure the quality of the measurements [27]. Another strength of this study lies in the methodology used to calculate the DSR scores. The available dietary intake data were disaggregated at the individual level into separate food products and ingredients. This enabled the derivation of species, allowing for a robust estimation of DSR over a two-day period for each participant. Therefore, the DSR scores in this study provide a reliable reflection of the dietary biodiversity of participants’ diets within the investigated food groups over the two measurement days. Lastly, the DHD15-index scores used in this study were previously calculated, checked and used in publications by the National Institute for Public Health and the Environment of the Netherlands, which ensures comparability of results across published studies [31,37].

This study also has some limitations. Firstly, misclassification of species occurs when examining species richness in diets. Previous publications describe a misclassification rate of 6–10% in ethnographic field research on species richness [38]. In this study, there were cases where insufficient information was available to determine a clear species for the food consumed. Thus, classification was based on assumptions in these cases, introducing a degree of uncertainty in the species classification of these foods. Nevertheless, this pertained to only a few specific food products and assumptions were based on conventional dietary patterns or local cuisine, where possible. Secondly, this study was based on two 24-h dietary recalls per participant. Therefore, it only captured DSR data over two measurement days. It is unclear if this adequately captures the variety of species consumed or if more measurement days are necessary. Other forms of dietary intake assessment (e.g., food-frequency questionnaires) would offer the ability to provide a broader overview of diets over longer periods of time. However, these methods often generate less detailed dietary intake data, complicating species derivation and raising the risk of misidentification and misclassification, which, in turn, can lead to reliable and robust DSR scores. Thirdly, the influence of seasonality in the association between DSR and DHD15 was not initially tested for. However, when adjusting for the season of dietary recall in additional regression analyses, it showed no substantial differences in the regression coefficients between DSR and the DHD15-index. Therefore, seasonality did not influence the associations between DSR and the DHD15-index.

Lastly, it is important to note that the overall DSR score did not reflect all food groups within the dietary pattern. This was because it became too challenging to accurately derive the correct species from the food registrations for these food products. These instances often concerned processed products consisting of a combination of species, making it unclear which recipe, thus species, to maintain. As highly processed products often comprise commonly consumed species (e.g., wheat, corn, cow, pork), we expected that excluding these products would have little impact on the DSR scores. Other excluded food groups accounted for <5% of the dietary pattern (oils, fats, condiments, herbs and spices). Thus, we anticipated that they would have minimal effect on the association between the DSR scores and the DHD15-index in this study. However, food groups that significantly contribute to biodiversity in the diet (fruit, vegetables, pulses and legumes, nuts and seeds) were included in overall DSR, along with minimally processed animal products [30].

Finally, we did not adjust for BMI as a proxy for energy intake due to selective missing BMI values in the ≥65 years age group (n = 516). Instead, we adjusted for total kilocalorie intake, which may not be preferable given that underreporting of kilocalorie intake is common in dietary intake studies and therefore could be considered as a limitation [39,40,41]. Nonetheless, sensitivity analysis with adjustment for BMI yielded similar results.

One important consideration is the challenge of disentangling the correlation between species richness and the amount of food consumed, either in grams or kilocalories. Efforts were undertaken to mitigate the impact of food quantity on the association between dietary species richness (DSR) and diet quality. Models were adjusted for total kilocalories and the quantity of consumed fruits and vegetables. Upon examination of the outcomes, it became apparent that grams of consumed fruits significantly weakened the association between DSR fruit and diet quality. DSR fruit also showed high correlations with quantity of consumed fruit in grams (cc > 0.7). Associations between DSR vegetables and diet quality became non-significant after adjustment for grams of consumed vegetables; however, DSR vegetables did not show high correlations with grams of consumed vegetables (cc < 0.4). It is plausible that the association between DSR and diet quality could be biased by the potential correlation between species richness and the amount of food consumed. This can be observed in the correlation coefficients, particularly for DSR in fruit and diet quality.

To our knowledge this is the first exploratory study to establish the positive association between dietary biodiversity and diet quality in Western populations. More research on the possible interplay among dietary biodiversity, diet quality and health outcomes in Western populations is warranted to elucidate how these factors relate to each other in Western dietary contexts. Future research could benefit from the use of species classification lists, developed and validated prior to obtaining dietary intake assessments, in order to prevent misclassification. Species classifications should be validated by experts in the area of local ecology and species with a preface on hybrid crops and cultivars. Recently, guidelines have been developed to accurately document species in food consumption studies [42]. Moreover, it is important that the methods used (e.g., dietary recall or food-frequency questionnaires) are sufficiently detailed and sensitive to distinguish between species in order to decrease the probability of misidentification and misclassification. Additionally, future research could investigate if dietary biodiversity is associated with other aspects of sustainability, such as biodiversity (cultivated and wild) and other environmental impact outcomes (e.g., greenhouse gas emissions, land use, water use). These findings can support recommendations to structurally incorporate aspects of biodiversity, one of the planetary boundaries currently transgressed, in national and international dietary guidelines, in the move towards more sustainable diets in line with the United Nations’ Sustainable Development Goals [10,22,43]. Current dietary guidelines advocate for a diverse diet to enhance health benefits. This study suggests that incorporating a more biodiverse diet could further improve diet quality and potentially contribute to better health outcomes. However, given the exploratory nature of these findings, it is premature to make definitive recommendations for the general population. Nonetheless, the results are consistent with existing dietary recommendations that emphasize the importance of a varied diet.

## 5. Conclusions

In conclusion, in this exploratory study, we found that higher dietary biodiversity positively contributes to the diet quality of Dutch adults. Therefore, DSR score could be considered as an extra component in evaluating diet quality in Dutch adults, in addition to other diversity scores or diet quality indicators. The results of this study also highlight the importance of further research on dietary biodiversity (species richness) and diet quality and, potentially, its impact on health outcomes.

## Figures and Tables

**Figure 1 nutrients-16-02189-f001:**
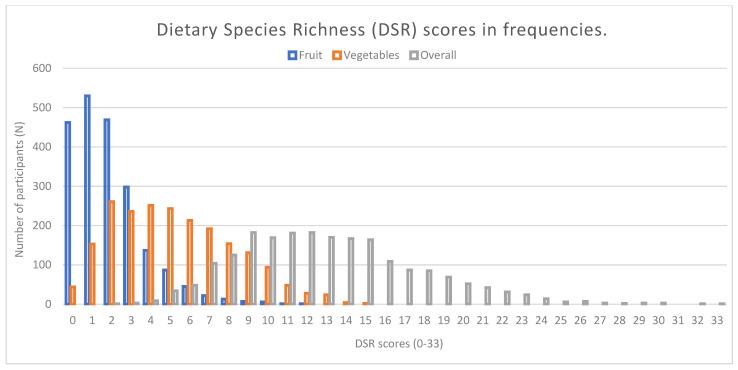
Frequencies chart of different dietary species richness (DSR) scores in number of participants (N) in the Dutch National Food Consumption Survey respondents (adults aged 19-79 years): DSR fruit (blue), DSR vegetables (orange) and overall DSR ^1^ (grey). ^1^ Included food groups: fruits, vegetables, nuts and seeds, legumes, tubers, grains, meat, fish and dairy.

**Table 1 nutrients-16-02189-t001:** Characteristics of the Dutch National Food Consumption Survey 2012–2016 respondents (adults aged 19–79 years), stratified by four age groups and sex (n = 2078).

	Total Population N = 2078		Age 19–30 Years N = 516		Age 31–50 Years N = 523		Age 51–64 Years N = 367		Age ≥65 Years N = 672	
	Men	Women	Men	Women	Men	Women	Men	Women	Men	Women
Sex, n (%)	1043 (50.3)	1035 (49.8)	260 (50.4)	256 (49.6)	259 (49.5)	264 (50.5)	178 (48.5)	189 (51.5)	346 (51.5)	326 (48.5)
Age (in years), M (SD)	50.8 (19.2)	50.2 (19.1)	25.3 (3.7)	25.1 (3.6)	42.1 (5.5)	41.2 (5.9)	58.1 (3.9)	58.0 (3.9)	72.8 (3.9)	72.6 (3.7)
Educational level, n (%)										
Low	242 (23.2)	360 (34.8)	27 (10.4)	28 (10.9)	39 (15.1)	53 (20.1)	49 (27.5)	73 (38.6)	127 (36.7)	206 (63.2)
Middle	406 (38.9)	383 (37.0)	116 (44.6)	106 (41.4)	114 (44.0)	133 (50.4)	68 (38.2)	76 (40.2)	108 (31.2)	68 (20.9)
High	395 (37.9)	292 (28.2)	117 (45.0)	122 (47.7)	106 (40.9)	78 (29.5)	61 (34.3)	40 (21.2)	111 (31.1)	52 (16.0)
BMI (in kg/m^2^), M (SD)	26.0 (4.6)	26.6 (5.6)	23.8 (3.8)	24.3 (4.8)	26.5 (4.4)	27.3 (5.5)	27.9 (4.7)	28.3 (6.1)	27.3 (4.4)	27.5 (4.3)
Energy intake (kcal/day), M (SD)	2489 (712)	1842 (487)	2673 (748)	1950 (538)	2637 (817)	1851 (539)	2541 (674)	1797 (468)	2213 (507)	1776 (388)
Quantity consumed (grams), M (SD) ^1^										
Fruits	239 (255)	280 (274)	204 (247)	265 (297)	202 (236)	231 (246)	246 (243)	275 (293)	290 (272)	335 (257)
Vegetables	301 (197)	298 (210)	300 (223)	266 (196)	269 (180)	273 (208)	289 (181)	310 (231)	333 (194)	356 (203)
Mean DSR scores, M(SD)										
DSR overall ^2^	12.9 (4.5)	13.1 (4.6)	12.7 (4.8)	12.9 (4.6)	12.7 (4.4)	12.5 (4.8)	12.8 (4.7)	13.9 (4.6)	13.1 (4.0)	13.4 (4.4)
DSR fruit	1.7 (1.7)	2.2 (1.9)	1.5 (1.7)	1.9 (1.9)	1.5 (1.6)	1.9 (1.7)	1.7 (1.7)	2.2 (1.9)	2.0 (1.7)	2.6 (1.9)
DSR vegetables	5.2 (3.0)	5.2 (3.0)	5.4 (3.2)	5.3 (2.9)	5.4 (3.1)	5.1 (3.2)	5.0 (3.0)	5.8 (3.2)	5.1 (2.8)	5.0 (2.8)
Median DSR scores, median (IQR)										
DSR overall ^2^	12 (6)	13 (6)	12 (6)	12 (7)	12 (6)	12 (7)	12 (7)	14 (6)	13 (5)	13 (6)
DSR fruit	1 (2)	2 (2)	1 (2)	1 (3)	1 (2)	2 (2)	1 (1)	2 (2)	2 (2)	2 (3)
DSR vegetables	5 (4)	5 (4)	5 (5)	5 (4)	5 (5)	5 (5)	5 (5)	5 (5)	5 (4)	4 (4)
DHD15-index score, M(SD) ^3^	53.7 (17.4)	65.1 (18.0)	50.6 (18.3)	62.5 (20.6)	51.6 (18.2)	63.6 (18.2)	52.3 (15.0)	65.1 (16.8)	58.2 (16.5)	68.4 (15.9)

N/n = number of participants, M = mean, SD = standard deviation of the mean, BMI = body mass index (kg/m^2^), kcal = kilocalories, DSR = dietary species richness, IQR = interquartile range. ^1^ Mean of sum of consumption (in grams) over two measurement days. ^2^ Included food groups: fruits, vegetables, nuts and seeds, legumes, tubers, grains, meat, fish and dairy. ^3^ Dutch Healthy Diet index 2015: 0 points indicates minimal adherence to dietary guidelines; 140 points indicates maximal adherence to dietary guidelines.

**Table 2 nutrients-16-02189-t002:** The associations between dietary biodiversity scores (DSR) and diet quality (DHD15-index score[min.0–max.140]) in Dutch National Food Consumption Survey 2012–2016 respondents (adults aged 19–79 years) stratified by age group.

**DSR Overall ᵒ**	**Total Population** **N = 2078**	**Age 19–30** **N = 516**	**Age 31–50** **N = 523**	**Age 51–64** **N = 367**	**Age ≥ 65** **N = 672**
Model 1 ^1^ B [95%-CI] *p*-value	1.28 [1.12–1.43] <0.001 *	1.50 [1.17–1.82] <0.001 *	0.93 [0.60–1.26] <0.001 *	1.18 [0.84–1.51] <0.001 *	1.08 [0.81–1.36] <0.001 *
Model 2 ^2^ B [95%-CI] *p*-value	1.40 [1.25–1.55] <0.001 *	1.62 [1.31–1.92] <0.001 *	1.10 [0.80–1.40] <0.001 *	1.28 [0.952–1.60] <0.001 *	1.24 [0.97–1.52] <0.001 *
**DSR fruit**	**Total population** N = 2078	**Age 19–30** N = 516	**Age 31–50** N = 523	**Age 51–64** N = 367	**Age ≥ 65** N = 672
Model 1 ^1^ B [95%-CI] *p*-value	3.96 [3.57–4.35] <0.001 *	3.99 [3.13–4.85] <0.001 *	3.54 [2.67–4.41] <0.001 *	3.64 [2.80–4.48] <0.001 *	3.45 [2.82–4.08] <0.001 *
Model 2 ^2^ B [95%-CI] *p*-value	4.01 [3.65–4.38] <0.001 *	4.28 [3.47–5.09] <0.001 *	3.65 [2.84–4.46] <0.001 *	3.62 [2.80–4.44] <0.001 *	3.69 [3.07–4.30] <0.001 *
Model 3 ^3^ B [95%-CI] *p*-value	1.91 [1.40–2.42] <0.001 *	2.50 [1.34–3.65] <0.001 *	0.87 [−0.21–1.94] 0.11	1.58 [0.50–2.66] 0.004 *	2.03 [1.21–2.86] <0.001 *
**DSR vegetables**	**Total population** N = 2078	**Age 19–30** N = 516	**Age 31–50** N = 523	**Age 51–64** N = 367	**Age ≥ 65** N = 672
Model 1 ^1^ B [95%-CI] *p*-value	0.79 [0.54–1.04] <0.001 *	1.26 [0.74–1.79] <0.001 *	0.74 [0.26–1.22] 0.002 *	0.74 [0.22–1.26] 0.005 *	0.25 [−0.18–0.67] 0.35
Model 2 ^2^ B [95%-CI] *p*-value	0.84 [0.61–1.08] <0.001 *	1.24 [0.74–1.74] <0.001 *	0.84 [0.39–1.29] <0.001 *	0.76 [0.25–1.27] 0.003 *	0.34 [−0.09–0.77] 0.116
Model 3 ^3^ B [95%-CI] *p*-value	0.16 [−0.07–0.40] 0.17	0.38 [−0.10–0.87] 0.12	0.30 [−0.16–0.76] 0.21	0.24 [−0.27–0.76] 0.36	−0.20 [−0.62–0.23] 0.36

N = number of participants, DSR = dietary species richness, B = unstandardized beta regression coefficient, 95% CI = 95% confidence interval. ᵒ Included food groups: *fruits, vegetables, nuts and seeds, legumes, tubers, grains, meat, fish and dairy*. * = *p*-value < 0.05. ^1^ adjusted for sex and educational level (reference group = low educational level). ^2^ adjusted for sex, educational level and mean energy intake (kcal/day). ^3^ adjusted for sex, educational level, mean energy intake (kcal/day) and quantity of consumed fruits or vegetables (in grams) over two measurement days.

## Data Availability

The data presented in this study are available on request from the corresponding author. DNFCS data are available for research. More information can be found on the DNFCS website: https://www.wateetnederland.nl/ (accessed on 30 April 2024).

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
