# Peer review of "Dietary Biodiversity and Diet Quality in Dutch Adults"

_nutrients, 2024, doi:10.3390/nu16142189_

Round 1
Reviewer 1 Report
Comments and Suggestions for Authors
I would like to thank the authors of this paper for putting together this interesting article. The abstract and the introduction were well-written with great insights of the relationship between dietary biodiversity and diet quality but the scope of the study in opinion is too narrow to convey a new and relevant finding to existing knowledge in this space. The authors stated that the study aims to investigate whether DSR and diet quality are associated in Dutch adults (aged 19-79 years)- used Diet Species Richness to measure dietary biodiversity and DDS to assess diet quality. Since data were for this study were retrieved from the Dutch National Food Consumption Survey (DNFCS) 2012-2016 carried out by the National Institute for Public Health and the Environment of the Netherlands, I think if more than one measure is used to assess dietary biodiversity and dietary quality in this study, it would improve the usability of their findings. Other measures of dietary biodiversity measures that can be used are Simpson’s index of diversity, which represents the number of different species consumed and how evenly the amounts consumed of these different species are distributed based on quantity consumed; and the functional diversity, as the total branch length of a functional dendrogram. These measures are subject to the data availability in the DNFCS. In addition to diet quality, nutritional quality indicators such as mean adequacy ratios and nutrient adequacy ratios of the Dutch diets can be included in the study. The authors can provide how DSR can be a proxy for diet and nutrition quality of foods consumed.
I suggest a paragraph on the implications of the findings and its applicability for a better nutritional outcome in the Western diets' context should be added in the discussion section.
Author Response
Dear reviewer,
In the attached word document you can see an overview of your comments and a point-by-point response formulated by the authors formulated (blue text).
Kind regards,

Reviewer 2 Report
Comments and Suggestions for Authors
I very much appreciate this paper and content I would like to suggest three aspects that are adding to the quality of your research:
a) give some reference, if there are differences that you could detect in the seasonality
b) participants in such surveys tend to have a bias to the "middle" and often miss out on less educated and low income groups. How about such issues in your survey?
c) what are practical recomendations and conclusions you might derive for the general Dutch population?
Author Response

(The authors gave the same response as above.)
